# ONLINE TUNING FOR OFFLINE DECENTRALIZED MULTI-AGENT REINFORCEMENT LEARNING

## ABSTRACT

Offline reinforcement learning could learn effective policies from a fixed dataset, which is promising in real-world applications. However, in offline decentralized multi-agent reinforcement learning, due to the discrepancy between the behavior policy and learned policy, the transition dynamics in offline experiences do not accord with the transition dynamics in online execution, which creates severe errors in value estimates, leading to uncoordinated and suboptimal policies. One way to overcome the transition bias is to bridge offline training and online tuning. However, considering both deployment efficiency and sample efficiency, we could only collect very limited online experiences, making it insufficient to use merely online data for updating the agent policy. To utilize both offline and online experiences to tune the policies of agents, we introduce *online transition correction* (OTC) to implicitly correct the biased transition dynamics by modifying sampling probabilities. We design two types of distances, *i.e.*, embedding-based and value-based distance, to measure the similarity between transitions, and further propose an adaptive rank-based prioritization to sample transitions according to the transition similarity. OTC is simple yet effective to increase data efficiency and improve agent policies in online tuning. Empirically, we show that OTC outperforms baselines in a variety of tasks.

## 1 INTRODUCTION

In fully decentralized multi-agent reinforcement learning (MARL) (de Witt et al., 2020a), agents interact with the environment to obtain individual experiences and independently improve the policies to maximize the cumulative shared reward. Due to the scalability, decentralized learning would be promising in real-world cooperative tasks. However, in many industrial applications, continuously interacting with the environment to collect the experiences for learning is costly and risky, *e.g.*, autonomous driving. To overcome this challenge, offline decentralized MARL (Jiang & Lu, 2021) lets each agent learn its policy from a fixed dataset of experiences without interacting with the environment. The dataset of each agent contains the individual action instead of the joint action of all agents. There is no assumption on the data collection policies and the relationship between the datasets of agents.

However, from the perspective of each individual agent, the transition dynamics depend on the policies of other agents and will change as other agents improve the policies (Foerster et al., 2017). Since the behavior policies of other agents during data collection would be inconsistent with their learned policies, the transition dynamics in the dataset would be different from the real transition dynamics in execution, which will cause extrapolation error, *i.e.*, the error in value estimate incurred by the mismatch between the experience distribution of the learned policy and the dataset (Fujimoto et al., 2019). The extrapolation error makes the agent underestimate or overestimate state values, which leads to uncoordinated and suboptimal policies (Jiang & Lu, 2021).

One way to reduce the extrapolation error caused by the mismatch of transition dynamics is to bridge offline training and online tuning. However, since both deploying new policies and interacting with the environment are costly and risky, we should consider the deployment efficiency (the number of deployments) and sample efficiency (the number of interactions) in the collection of online experiences (Matsushima et al., 2021). Due to the efficiency requirement, the collected online experiences can be very limited. Thus, it is insufficient to tune the policies of agents merely using the online

data, and the small online dataset may also cause overfitting. To increase data efficiency, it is better to additionally exploit the offline data for online tuning. However, uniformly sampling from the merged offline and online experiences (Nair et al., 2020) cannot address the transition mismatch problem. Therefore, it is necessary to correct the transition dynamics in the offline data to make it close to the online transition dynamics. Nevertheless, the requirement of efficiency also means it is infeasible to accurately estimate the real transition dynamics from the limited online experiences, thus explicit correction is impractical.

In this paper, we introduce a simple yet effective method to correct the transition dynamics of offline data for online tuning, without explicitly modeling the transition dynamics. When sampling a transition from the offline experiences, we first search for the best-matched transition in the online experiences, which has the minimum state-action distance to the sampled transition. Then, we compute the next-state distance between the sampled transition and the best-matched online transition to represent the transition similarity. After that, a probability function maps the transition similarity to the probability of accepting the sampled transition for update, which is equivalent to modifying the transition probability. Therefore, the objective is to find the optimal probability function which minimizes the KL-divergence between the online transition distribution and the modified transition distribution, given the distance measure. We design two distance measures based on the embedding and Q-value of transitions, respectively. The embedding-based distance captures the similarity in feature space, and the value-based distance measures the isomorphism. Due to the limited online experiences, it is hard to find the optimal probability function by gradient-based optimization, but we empirically find that the rank-based prioritization in PER (Schaul et al., 2016) is a proper choice of the probability function. Moreover, we propose an adaptive rank-based prioritization to adjust the degree of the correction according to the difference between offline and online transition distributions.

The proposed method, termed OTC (**O**nline **T**ransition **C**orrection), could be applied to any offline RL method for decentralized multi-agent learning. We construct the decentralized datasets from a variety of D4RL tasks (Fu et al., 2020) and evaluate OTC on them. Experimental results show that OTC outperforms baselines, and ablation studies demonstrate the effectiveness of the two distance measures, the practicability of rank-based prioritization, and the improvement of adaptive prioritization. To the best of our knowledge, OTC is the first method for bridging offline training and online tuning in decentralized MARL.

## 2 RELATED WORK

### 2.1 OFFLINE RL

In offline RL, the agent could only access to a fixed dateset of single-step transitions collected by a behavior policy, and no interactive experience collection is allowed during learning. Offline RL easily suffers from the extrapolation error, which is mainly caused by out-of-distribution actions in single-agent environments. To address this issue, constraint-based methods introduce policy constraints to enforce the learned policy to be close to the behavior policy, *e.g.*, direct action constraint (Fujimoto et al., 2019), kernel MMD (Kumar et al., 2019), Wasserstein distance (Wu et al., 2019), and KL-divergence (Peng et al., 2019). Conservative methods (Kumar et al., 2020; He & Hou, 2020; Yu et al., 2021) train a Q-function pessimistic to out-of-distribution actions. Uncertainty-based methods quantify the uncertainty by the learned environment model (Yu et al., 2020) or by Monte Carlo dropout (Wu et al., 2021) of Q-function, and use it as a penalty or to weight the update of Q-function, so as to avoid the overestimation of out-of-distribution actions.

In offline decentralized MARL, besides out-of-distribution actions, the extrapolation error is also caused by the bias of transition dynamics. For each individual agent, since the transition dynamics depend on other agents' policies which are also updating, there will be a difference between the transition dynamics in the offline dataset and the real transition dynamics during online deployment (Jiang & Lu, 2021). To overcome this, MABCQ (Jiang & Lu, 2021) uses two importance weights to modify the offline transition dynamics by normalizing the biased transition probabilities and increasing the transition probabilities of high-value next states. However, the modified transition dynamics in MABCQ are not theoretically guaranteed to be close to the real ones. Unlike MABCQ, OTC focuses on online tuning and exploits online experiences to correct the bias of transition dynamics to quickly adapt to the learned policies of other agents.

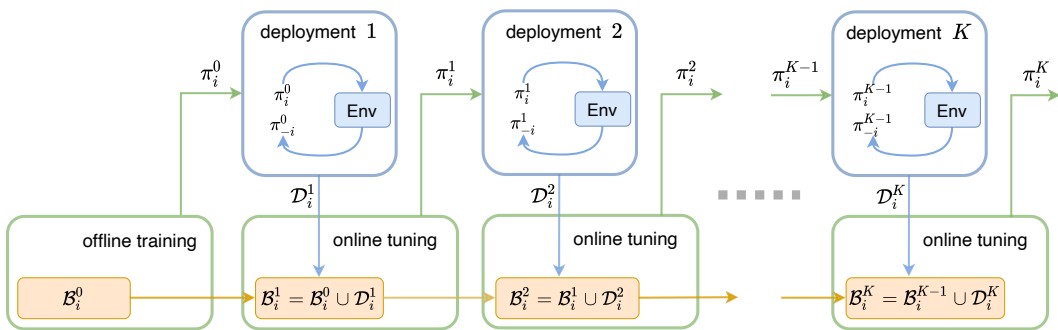

Figure 1: Overview of offline training and online tuning. After each agent $i$ learning its policy $\pi_i^0$ from offline dataset $\mathcal{B}_i^0$, their policies are deployed in the environment to get the online dataset $\mathcal{D}_i^1$. Then, $\pi_i^0$ is finetuned to obtain $\pi_i^1$ using the merged dataset $\mathcal{B}_i^1$. The online tuning is repeated for $K$ times.

## 2.2 BRIDGING OFFLINE LEARNING AND ONLINE TUNING

Since the offline dataset is usually insufficient to cover the entire transition space, the extrapolation error cannot be eliminated entirely in the fully offline learning. It is crucial to improve the policy trained using offline data further with online reinforcement learning. Since the online interaction is expensive, we must consider both the deployment efficiency (the number of policy deployments) and sample efficiency (the number of interactions) in online tuning. The concept of deployment efficiency is adopted in BREMEN (Matsushima et al., 2021) and MUSBO (Su et al., 2021), which, however, do not aim to finetune the pre-trained policy but instead train the policy from scratch with limited deployments. AWAC (Nair et al., 2020) employs an implicit constraint that could mitigate the extrapolation error while avoiding overly conservative updates in offline learning and thus quickly performs online finetuning. Balanced Replay (Lee et al., 2021) adopts prioritized sampling to encourage the use of near-on-policy samples from the offline dataset. However, they deploy the policy frequently, ignoring the deployment efficiency. Moreover, these methods above are designed for single-agent environments, where offline and online data follow the same transition dynamics, thus they cannot deal with transition bias. Abiding by both deployment and sample efficiency, OTC uses prioritized sampling to reduce the bias of transition dynamics in decentralized MARL, rather than the state-action distribution shift considered in Balanced Replay.

## 3 METHOD

### 3.1 PRELIMINARIES

In offline and decentralized cooperative settings, each agent $i$ could only access to an offline dataset $\mathcal{B}_i$, which is collected by a behavior policy and contains the tuples $\langle s, a_i, r, s' \rangle$, where $s$ is the state, $a_i$ is the individual action of agent $i$, $r$ is the shared reward, and $s'$ is the next state. Note that $\mathcal{B}_i$ dose not contain the joint actions of all agents. Each agent $i$ independently learns its policy $\pi_i$ using offline RL algorithm, without information sharing among agents. The goal of all agents is to maximize the expected return $\mathbb{E} \sum_{t=0}^T \gamma^t r_t$ when deploying their learned policies in the environment, where $\gamma$ is the discount factor and $T$ is the time horizon of the episode. From the perspective of each agent $i$, the transition probability in $\mathcal{B}_i$, denoted by $P_{\mathcal{B}_i}(s'|s, a_i)$, depends on other agents' behavior policies during the collection of $\mathcal{B}_i$, while the real transition probability in execution, denoted by $P_{\mathcal{E}_i}(s'|s, a_i)$, depends on other agents' learned policies. The difference between $P_{\mathcal{B}_i}$ and $P_{\mathcal{E}_i}$ would cause severe extrapolation errors, which eventually lead to uncoordinated and suboptimal policies (Jiang & Lu, 2021).

To finetune the policies learned from offline datasets, we allow the agents to interact with the environment to collect online experiences. As illustrated in Figure 1, after offline learning using initial dataset $\mathcal{B}_i^0$ for each agent $i$, their learned policies $\langle \pi_i^0, \pi_{-i}^0 \rangle$, where $\pi_{-i}^0$ denotes the policies of all agents except $i$, are deployed in the environment and interact with each other for $M$ timesteps. Each agent $i$ obtains an online dataset $\mathcal{D}_i^1$ with $M$ transitions. $\mathcal{D}_i^1$ still only contains the individual actions of agent $i$ rather than the joint actions. We merge $\mathcal{D}_i^1$ and $\mathcal{B}_i^0$ to get $\mathcal{B}_i^1$, and finetune the policy $\pi_i^0$ to obtain $\pi_i^1$ using $\mathcal{B}_i^1$. Then, we deploy the updated policies $\langle \pi_i^1, \pi_{-i}^1 \rangle$ in the environment and repeat

the procedures until $K$ deployments. $K$ represents the deployment efficiency, and $K \times M$ represents the sample efficiency. Note that for presentation simplicity, we denote $\pi_{-i}$ as also updating, but other agents may or may not be learning online, which however does not affect the following problem formulation and our method. That said OTC does not have any assumptions on other agents.

## 3.2 PROBLEM FORMULATION

For agent $i$, given $s$ and $a_i$, the KL-divergence between the transition distributions of next state $s'$ in the online dataset $\mathcal{D}_i^k$ and in the merged dataset $\mathcal{B}_i^k$ is

$$\text{KL}(P_{\mathcal{D}_i^k} \| P_{\mathcal{B}_i^k}) = \sum_{s'} P_{\mathcal{D}_i^k}(s'|s, a_i) \log \frac{P_{\mathcal{D}_i^k}(s'|s, a_i)}{P_{\mathcal{B}_i^k}(s'|s, a_i)}. \tag{1}$$

However, since $\text{KL}(P_{\mathcal{D}_i^k} \| P_{\mathcal{B}_i^k})$ is generally large, in order to use the merged dataset to finetune the policy, we need to modify $P_{\mathcal{B}_i^k}$ as $\tilde{P}_{\mathcal{B}_i^k}$ to minimize the KL-divergence between $P_{\mathcal{D}_i^k}$ and $\tilde{P}_{\mathcal{B}_i^k}$.

Since the difference of transition distributions means the difference of next-state distributions given the same state-action pair, we first define two distance functions: $d(s_1, a_{i_1}, s_2, a_{i_2})$ that measures the similarity of state-action pairs, and $d(s_1', s_2')$ that measures the similarity of next states. Once sampling a transition $\langle s, a_i, s', r \rangle$ from $\mathcal{B}_i^k$, we select the best-matched transition $\langle s^*, a_i^*, s'^*, r^* \rangle$ from $\mathcal{D}_i^k$, which has the minimum state-action distance to $\langle s, a_i, s', r \rangle$,

$$\langle s^*, a_i^*, s'^*, r^* \rangle = \arg \min_{\mathcal{D}_i^k} d(s, a_i, s^*, a_i^*). \tag{2}$$

For the convenience of theoretical analysis, we assume there is always $\langle s^*, a_i^*, s'^*, r^* \rangle$ in $\mathcal{D}_i^k$ that meets $d(s, a_i, s^*, a_i^*) = 0$, *i.e.*, $s = s^*$ and $a_i = a_i^*$. If there is more than one transition, we uniformly select one from them. Then, we adopt a probability function $f$ which maps the distance $d(s', s'^*)$ to the probability of accepting the sampled transition $\langle s, a_i, s', r \rangle$ for update, *i.e.*, $f(d(s', s'^*))$. Therefore, the transition probability can be modified as

$$\tilde{P}_{\mathcal{B}_i^k}(s'|s, a_i) = P_{\mathcal{B}_i^k}(s'|s, a_i) * \frac{\sum_{\hat{s}'} P_{\mathcal{D}_i^k}(\hat{s}'|s, a_i) f(d(s', \hat{s}'))}{Z(s, a_i)}, \tag{3}$$

where $Z(s, a_i)$ is a normalization term to make sure $\sum_{s'} \tilde{P}_{\mathcal{B}_i^k}(s'|s, a_i) = 1$. Thus the KL-divergence between $P_{\mathcal{D}_i^k}$ and $\tilde{P}_{\mathcal{B}_i^k}$ is

$$\text{KL}(P_{\mathcal{D}_i^k} \| \tilde{P}_{\mathcal{B}_i^k}) = \text{KL}(P_{\mathcal{D}_i^k} \| P_{\mathcal{B}_i^k}) - \sum_{s'} P_{\mathcal{D}_i^k}(s'|s, a_i) \log \frac{\sum_{\hat{s}'} P_{\mathcal{D}_i^k}(\hat{s}'|s, a_i) f(d(s', \hat{s}'))}{Z(s, a_i)}. \tag{4}$$

To minimize the KL-divergence, we need to design appropriate $d$-functions to accurately measure the distances between transitions and find the optimal $f$-function which properly satisfies $\max_f \sum_{s'} P_{\mathcal{D}_i^k}(s'|s, a_i) \log(\sum_{\hat{s}'} P_{\mathcal{D}_i^k}(\hat{s}'|s, a_i) f(d(s', \hat{s}')) / Z(s, a_i))$.

## 3.3 $d$-FUNCTIONS

Due to the limitations of computational complexity and representation ability, directly measuring the distances in the original space is impractical, especially in high-dimensional environments. Therefore, we design two types of $d$-functions. The first type is the distance in the embedding space. We employ VAE (Kingma & Welling, 2013) to encode the state-action pair and next state into $e(s, a_i)$ and $e(s')$, and define the $d$-functions as $l1$ distance in the embedding space,

$$d_e(s, a_i, s^*, a_i^*) = \|e(s, a_i) - e(s^*, a_i^*)\|, \quad d_e(s', s'^*) = \|e(s') - e(s'^*)\|. \tag{5}$$

Due to the requirement of sample efficiency, it is impossible that we could always find the transition from $\mathcal{D}_i$ with the same state-action pair as the given transition. Relying on the generalization ability of the encoder, similar inputs will be encoded into similar embeddings. We could search for the best-matched transition with the most similar state-action pair in terms of $d_e(s, a_i, s^*, a_i^*)$ and then evaluate the transition similarity using $d_e(s', s'^*)$.

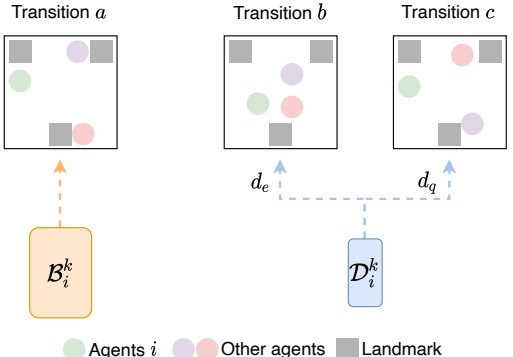

Figure 2: In this navigation task, the agents are learning to cover all the landmarks. For the transition $a$ from $\mathcal{B}_i^k$, the best-matched transition $b$ selected from $\mathcal{D}_i^k$ using $d_e$ is still much different from transition $a$. However, the value-based distance $d_q$ will select the nearly isomorphic transition $c$, which is more helpful for evaluating the trainsition similarity.

The embedding-based distance measures the similarity in feature space. However, since the limited online experiences cannot cover all state-action pairs in offline dataset, there must be some transitions in the offline dataset which are still much different from the transition with the most similar state-action feature, *e.g.*, transition $a$ and $b$ in Figure 2. In such cases, we cannot accurately evaluate the transition similarity using $d_e$. Inspired by state representation learning (Gelada et al., 2019; Zhang et al., 2020a;b), we notice that some state-action pairs may have common latent structure, *e.g.*, transition $a$ and $c$ in Figure 2. Although they are different in feature space, the topologies of agents are isomorphic. As pointed by DeepMDP (Gelada et al., 2019), nearly isomorphic state-action pairs would have similar Q-values. Therefore, we use the difference in Q-values to evaluate the isomorphism in state-action pairs and select the best-matched transition from the online dataset,

$$d_q(s, a_i, s^*, a_i^*) = \|Q(s, a_i) - Q(s^*, a_i^*)\|. \tag{6}$$

On the other hand, we use the expected Q-value to measure the distance of next states,

$$d_q(s', s'^*) = \left\|\frac{V(s')}{\mathbb{E}_{s'} V(s')} - \frac{V(s'^*)}{\mathbb{E}_{s'^*} V(s'^*)})\right\|, \quad \text{where } V(s') = \mathbb{E}_{a_i} Q(s', a_i). \tag{7}$$

$V(s')/\mathbb{E}_{s'} V(s')$ is the deviation from the expected value, which could mitigate the influence of absolute value. The value-based distance $d_q$ has stronger representation ability than the embedding-based distance $d_e$. The transitions that are close in embedding space will also have small value difference, and the value-based distance could also represent the isomorphism of transitions.

### 3.4 $f$-FUNCTION

The optimal $f$-function is to maximize $\sum_{s'} P_{\mathcal{D}_i^k}(s'|s, a_i) \log \sum_{\hat{s}'} P_{\mathcal{D}_i^k}(\hat{s}'|s, a_i) f(d(s', \hat{s}'))/Z(s, a_i)$. However, since the online experiences are limited, it is hard to solve the optimal $f$-function by gradient-based optimization. Nevertheless, there must exist such $f$-functions which do not increase $\text{KL}(P_{\mathcal{D}_i^k}\|\tilde{P}_{\mathcal{B}_i^k})$, and a trivial example is the constant function. Therefore, we try to find a heuristic and practical $f$-function which is able to reduce the KL-divergence and extrapolation error. An important prior is that $f$-function should be monotonic, which will produce a larger acceptance probability when fed with a smaller distance of next states. The intuition is that if the next state of the transition from $\mathcal{B}_i^k$ is more similar to the next state of the online experience with the same state-action pair, the transition is more likely to follow the transition dynamics in $\mathcal{D}_i^k$. Therefore, we should give it a larger acceptance probability, which eventually leads to a larger transition probability. Empirically, we find the rank-based prioritized sampling in PER (Schaul et al., 2016) is a good solution. Concretely, the probability of accepting transition $j$ is

$$P(j) = \frac{p_j^\alpha}{\sum_m p_m^\alpha}, \tag{8}$$

where the priority $p_j = 1/\text{rank}(j)$, and $\text{rank}(j)$ is the rank of transition $j$ when the transitions are sorted according to $d(s', s'^*)$. The exponent $\alpha$ determines the degree of modifying the transition

---

**Algorithm 1** OTC for Agent $i$

---

1: Initialize the RL model and the modification degree $\alpha_i^0$
2: Train the RL model using $\mathcal{B}_i^0$ to obtain the policy $\pi_i^0$
3: **for** $k = 1, \ldots, K$ **do**
4:     Deploy the policy $\pi_i^{k-1}$ in the environment
5:     Collect the online dataset $\mathcal{D}_i^k$ with $M$ transitions
6:     Merge the experiences $\mathcal{B}_i^k = \mathcal{B}_i^{k-1} \cup \mathcal{D}_i^k$
7:     **if** $k > 1$ **then**
8:         Adjust $\alpha_i^k$ by (9)
9:     **end if**
10:     **for** $t = 1, \ldots, max\_update$ **do**
11:         Sample a minibatch B from $\mathcal{B}_i^k$ and a minibatch D from $\mathcal{D}_i^k$
12:         **for** each transition in B **do**
13:            find the best-matched transition in D (2) and compute the transition similarity
14:         **end for**
15:         Sample transitions from B by rank-based prioritization (8)
16:         Update the RL model using the sampled transitions
17:     **end for**
18: **end for**

---

probability, with $\alpha = 0$ meaning the $f$-function degrades into a constant function. The rank-based prioritization ensures that the probability of being accepted is monotonic in terms of transition similarity, and is robust as it is insensitive to outliers.

Another prior is that the modification degree should depend on the distribution difference between $\mathcal{D}_i^k$ and $\mathcal{B}_i^k$. We should adopt a weaker modification degree when the transition dynamics in $\mathcal{B}_i^k$ is more similar to that in $\mathcal{D}_i^k$, and vice versa. Inspired by that, we propose to adaptively adjust $\alpha$ at each deployment $k$ ($k > 1$) as

$$\alpha_i^k = \alpha_i^{k-1} \times \frac{\mathbb{D}_i^k}{\mathbb{D}_i^{k-1}}, \quad \text{where } \mathbb{D}_i^k = \mathbb{E}_{s,a,s' \sim \mathcal{B}_i^k} \, d(s', s'^* \sim \underset{\mathcal{D}_i^k}{\arg\min} \, d(s, a_i, s^*, a_i^*)). \quad (9)$$

$\mathbb{D}_i^k$ is the expected transition similarity for agent $i$ at deployment $k$. As the difference between offline and online transition distributions would change along with the update of agents, a fixed $\alpha$ is not an optimal solution. On the contrary, for example, if the distribution difference grows, adaptive $\alpha$ is likely to take on large value, and thus more aggressively modifies the transition dynamics.

### 3.5 IMPLEMENTATION DETAILS

For the embedding-based distance $d_e$, we do not maintain two embeddings $e(s, a_i)$ and $e(s')$, but train a conditional VAE $G_i = \{E_i(\mu, \sigma | s, a_i, s'), D_i(a_i | s, s', z \sim (\mu, \sigma))\}$ which encodes $\langle s, a_i, s' \rangle$ into the embedding $\mu(s, a_i, s')$. We take the embedding $\mu(s, a_i, s')$ as a substitute for both $e(s, a_i)$ and $e(s')$, which is more effective and computationally efficient in practice. Moreover, it is costly to sample transitions from $\mathcal{B}_i^k$ and then search the best-match transitions from $\mathcal{D}_i^k$ for every update. To reduce the complexity, for each update we uniformly sample two minibatches, B and D respectively from $\mathcal{B}_i^k$ and $\mathcal{D}_i^k$, and perform the rank-based prioritized sampling on the two minibatches. The complete training procedure is summarized in Algorithm 1.

## 4 EXPERIMENTS

### 4.1 SETTINGS

We evaluate OTC in D4RL (Fu et al., 2020) datasets with three types: random, medium, and medium-replay. Following the settings in multi-agent mujoco (de Witt et al., 2020b; Jiang & Lu, 2021), we split the original action space of three mujoco tasks (Todorov et al., 2012), *i.e.*, HalfCheetah, Walker2d, and Hopper, into several sub-spaces. As illustrated in Figure 3, different colors indicate different agents. Each agent obtains the state and reward of the robot and independently

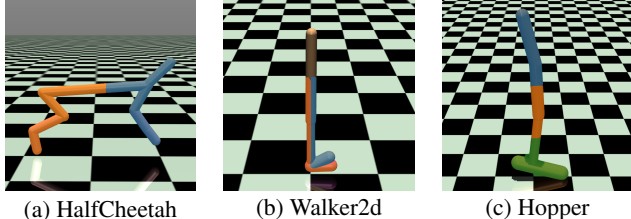

| (a) HalfCheetah | (b) Walker2d | (c) Hopper |

Figure 3: Illustrations of multi-agent mujoco tasks. Different colors mean different agents (Jiang & Lu, 2021).

controls one or some joints of the robot. For each agent $i$, we delete the actions of other agents from the original dataset and take the modified dataset as $\mathcal{B}_i^0$. During online tuning, we perform $K = 10$ deployments. For each deployment $k$, the agent collects a very limited online dataset $\mathcal{D}_i^k$, of which the size is only one percent of the initial offline dataset ($|\mathcal{D}_i^k| = 1\%|\mathcal{B}_i^0|$). After online data collection, we finetune the agents by $L$ updates and deploy the updated policies in the environment for next deployment.

We instantiate OTC respectively on two offline RL algorithms, BCQ (Fujimoto et al., 2019) and AWAC (Nair et al., 2020), and also take them as the baselines. During online tuning, the baselines uniformly and randomly sample transitions from the merged offline and online dataset, and they also have the same neural network architectures and hyperparameters as OTC. All the models are trained for five runs with different random seeds, and results are presented using mean and standard deviation. More details about hyperparameters are available in Appendix A.

## 4.2 Evaluating $d$-Functions

We summarize the performance of the last deployment of OTC with $d_e$ (embedding-based distance) and $d_q$ (value-based distance) on BCQ in Table 1 and on AWAC in Table 2, and plot the learning curves along with the number of deployments for a part of tasks in Figure 4. The empirical results show that OTC with $d_e$ or $d_q$ performs more than one standard deviation better than BCQ and AWAC, which verifies that $d_e$ and $d_q$ are capable of properly measuring the transition similarity. Since the online experiences are limited, uniformly sampling from the merged dataset is not effective

Table 1: Performance of OTC on BCQ.

|  | $d_e$ + BCQ | $d_q$ + BCQ | BCQ |
| --- | --- | --- | --- |
| halfcheetah-random | **1242** $\pm$ 11 | 1170 $\pm$ 16 | 1078 $\pm$ 23 |
| walker2d-random | **446** $\pm$ 24 | 444 $\pm$ 24 | 374 $\pm$ 32 |
| hopper-random | **328** $\pm$ 4 | 325 $\pm$ 8 | 309 $\pm$ 39 |
| halfcheetah-medium-replay | **2828** $\pm$ 65 | 2724 $\pm$ 35 | 2624 $\pm$ 55 |
| walker2d-medium-replay | 634 $\pm$ 40 | **730** $\pm$ 72 | 581 $\pm$ 128 |
| hopper-medium-replay | 753 $\pm$ 63 | **777** $\pm$ 271 | 568 $\pm$ 22 |
| halfcheetah-medium | 3725 $\pm$ 134 | **3732** $\pm$ 34 | 3638 $\pm$ 97 |
| walker2d-medium | **1449** $\pm$ 247 | 1406 $\pm$ 89 | 982 $\pm$ 49 |
| hopper-medium | 1286 $\pm$ 47 | **1405** $\pm$ 33 | 1169 $\pm$ 87 |

Table 2: Performance of OTC on AWAC.

|  | $d_e$ + AWAC | $d_q$ + AWAC | AWAC |
| --- | --- | --- | --- |
| halfcheetah-random | 296 $\pm$ 276 | **301** $\pm$ 243 | 59 $\pm$ 110 |
| walker2d-random | **660** $\pm$ 270 | 278 $\pm$ 21 | 262 $\pm$ 8 |
| hopper-random | **552** $\pm$ 79 | 459 $\pm$ 65 | 261 $\pm$ 61 |
| halfcheetah-medium-replay | 2990 $\pm$ 216 | **3263** $\pm$ 97 | 2578 $\pm$ 188 |
| walker2d-medium-replay | 409 $\pm$ 67 | **459** $\pm$ 37 | 368 $\pm$ 60 |
| hopper-medium-replay | **2943** $\pm$ 130 | 1877 $\pm$ 703 | 1741 $\pm$ 455 |
| halfcheetah-medium | **4253** $\pm$ 56 | 4176 $\pm$ 115 | 4090 $\pm$ 76 |
| walker2d-medium | 2027 $\pm$ 310 | **2099** $\pm$ 588 | 1059 $\pm$ 673 |
| hopper-medium | **2561** $\pm$ 533 | 2275 $\pm$ 785 | 1403 $\pm$ 384 |

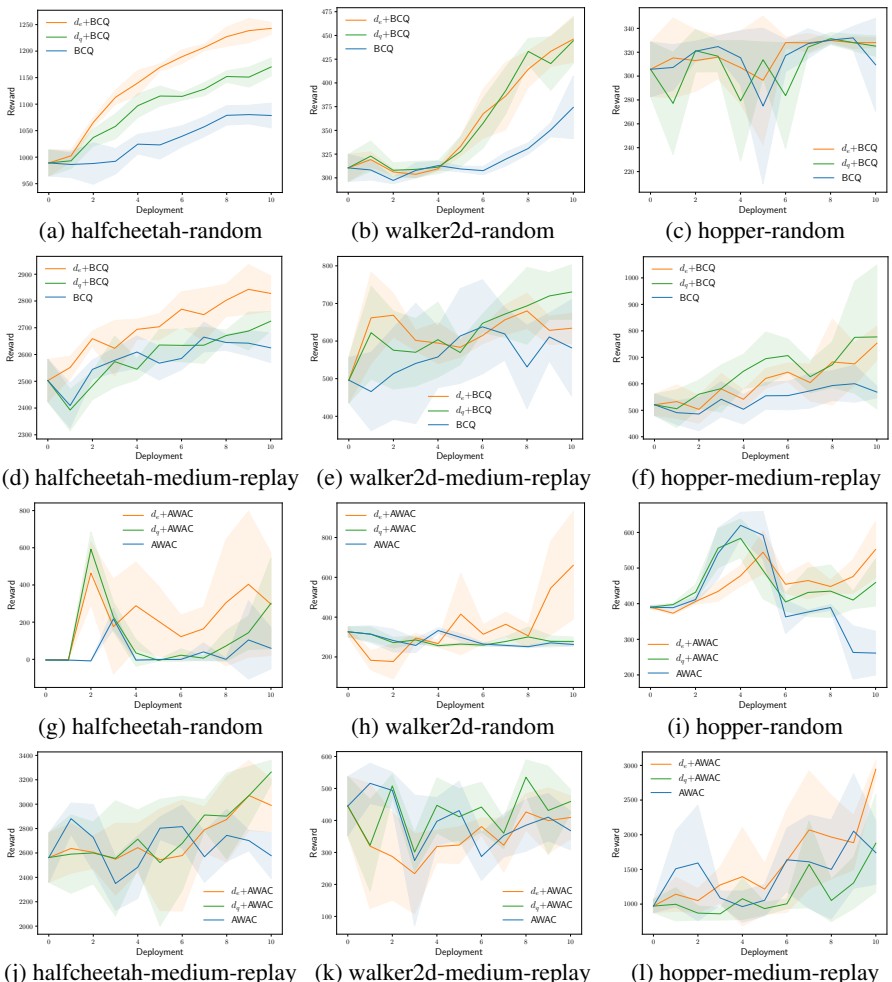

Figure 4: Learning curves of OTC on BCQ and AWAC.

in correcting the transition bias. The value-based distance $d_q$ has a stronger representation ability than the embedding-based distance $d_e$, since it could represent both the similarity in feature and isomorphism. However, $d_q$ does not commonly outperform $d_e$. The reason might be that $d_q$ would mistakenly judge the state-action pairs, which are different in feature and isomorphism but have similar values, as best-matched pairs. Moreover, since the Q-values are updating, $d_q$ is inconsistent during the tuning process.

### 4.3 EVALUATING $f$-FUNCTION

The hyperparameter $\alpha$ controls the strength of modifying the transition dynamics. Figure 5 shows the learning curves of OTC ($d_e$) with different $\alpha$. It is observed that if $\alpha$ is too small, OTC has weak effects on correcting the transition dynamics. However, if $\alpha$ is too large, the overly modified transition dynamics would deviate from the real transition dynamics and degrade the performance. Since the agents are continuously updated, the difference between the transition dynamics in $\mathcal{B}_i^k$ and $\mathcal{D}_i^k$ will also change every deployment, so a fixed $\alpha$ cannot deal with the nonstationarity. We test the fixed $\alpha = 1.0$ and our method for adaptive $\alpha$ where initial $\alpha^0 = 1.0$. Figure 6 shows that adaptive $\alpha$ could improve the performance of online tuning. Figure 7 shows the curves of $\alpha$ during the online tuning of OTC ($d_e$). In halfcheetah-medium-replay, $\alpha$ of both two agents grows, which means the difference between $\mathcal{B}_i^k$ and $\mathcal{D}_i^k$ increases as the agents are further improved during online tuning, and $\alpha$ becomes stable in the latter deployments, which means the convergence of online tuning. In halfcheetah-random and walker2d-random, two agents have different trends of $\alpha$, which means that the agents are influenced differently by the update of other agent. Our adaptive $\alpha$ method could discriminate the transition biases of agents rather than treat them equally.

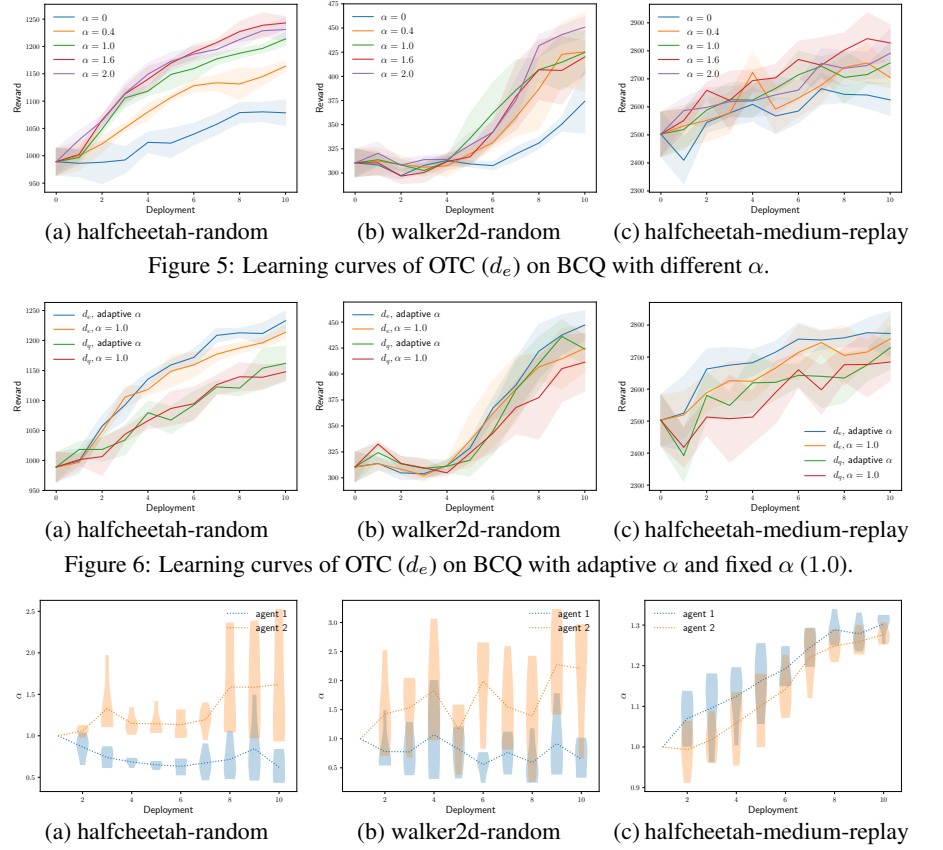

Figure 5: Learning curves of OTC ($d_e$) on BCQ with different $\alpha$.

(a) halfcheetah-random     (b) walker2d-random     (c) halfcheetah-medium-replay

Figure 6: Learning curves of OTC ($d_e$) on BCQ with adaptive $\alpha$ and fixed $\alpha$ (1.0).

(a) halfcheetah-random     (b) walker2d-random     (c) halfcheetah-medium-replay

Figure 7: Curves of adaptive $\alpha$ of OTC ($d_e$) on BCQ during online tuning. Dotted lines show mean values, and violin plots show distributions over seeds.

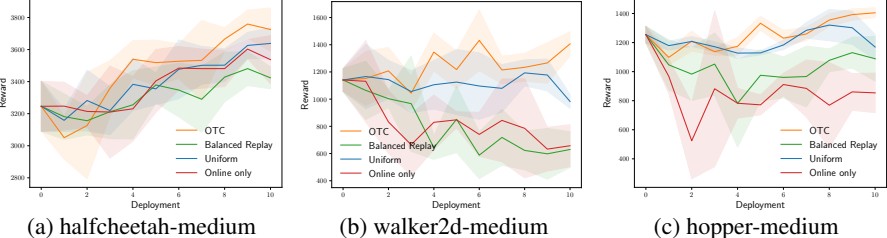

(a) halfcheetah-medium     (b) walker2d-medium     (c) hopper-medium

Figure 8: Learning curves of OTC, Balanced Replay, uniformly sampling from the merged dataset, and only sampling from the online dataset.

We additionally provide ablation studies of different sampling strategies in Figure 8. Since the online dataset is very limited, finetuning the agents with online samples only is insufficient, and the small dataset could cause overfitting, *e.g.*, Figure 8(b) and 8(c). Balanced Replay (Lee et al., 2021) and uniformly sampling are susceptible to the transition bias. OTC obtains the performance gain over the other three sampling strategies.

## 5   CONCLUSION

We have proposed OTC to effectively correct the transition dynamics during online interaction for tuning decentralized multi-agent policies learned from offline datasets, given limited online experiences. OTC consists of two types of distances to measure the transition similarity and an adaptive rank-based prioritization to sample transitions for updating the agent policy according to the transition similarity. Experimental results show that OTC outperform baselines for online tuning in a variety of tasks.

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

## A HYPERPARAMETERS

The hyperparameters are summarized in Table 3. For the results in Table 1 and Table 2, we use grid search to find the optimal $\alpha$ from $[0.6, 0.8, 1.0, 1.2, 1.4, 1.6]$.

Table 3: Experimental settings and hyperparameters

| Hyperparameter | BCQ | AWAC |
|---|---|---|
| discount ($\gamma$) | 0.99 | 0.99 |
| $\|B\|$ | 512 | 512 |
| $\|D\|$ | 2000 | 2000 |
| batch size | 128 | 128 |
| hidden sizes | $(64, 64)$ | $(256, 256)$ |
| activation | ReLU | ReLU |
| actor learning rate | $10^{-4}$ | $10^{-4}$ |
| critic learning rate | $10^{-4}$ | $5 \times 10^{-4}$ |
| embedding dimension | 10 | 10 |
| finetuning updates ($L$) | 4000 | 2000 |

## B OTC ON MABCQ

As the transition distribution of the learned policy in MABCQ does not follow $P_{\mathcal{B}_i}$, MABCQ is not a suitable backbone algorithm, though we additionally provides some results of OTC on MABCQ in Figure 9. OTC+MABCQ could outperform MABCQ in online tuning. In halfcheetah-random, MABCQ achieves better performance than BCQ in offline learning. However, since value deviation in MABCQ has made the agent be optimistic toward other agents in offline training, the modified transition dynamics in MABCQ is close to the real transition dynamics during the online interaction with improved other agents, and thus MABCQ does not benefit from online tuning in halfcheetah-random.

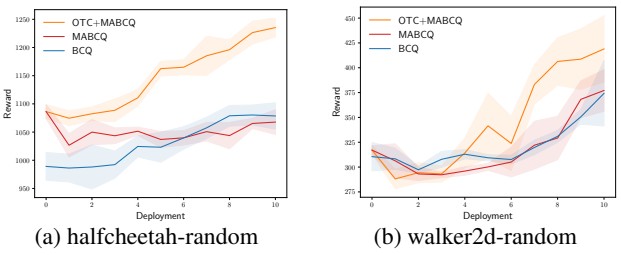

(a) halfcheetah-random      (b) walker2d-random

Figure 9: Learning curves of OTC ($d_e$) on MABCQ.

