# OpenReview forum: "Online Tuning for Offline Decentralized Multi-Agent Reinforcement Learning"
_ICLR.cc/2022/Conference — ICLR 2022 Submitted_

### Official Review · Reviewer_ZgL6 · 2021-10-18

**Correctness:** 3
**Technical Novelty And Significance:** 2
**Empirical Novelty And Significance:** 2
**Recommendation:** 3
**Confidence:** 5

**Details Of Ethics Concerns:**

no.

**Main Review:**

Overall, I don't have issues with the technical proposal of this paper.

* The author studied an important problem in offline MARL. When the opponent agents' actions are not accurate, it can cause mismatch between the transition dynamics in the offline data and the transition dynamics in real online settings. As a remedy, the authors proposed to solve this issue by bridging offline training with online tuning.


* OTC essentially tries to find the optimal probability function that minimizes the KL divergence between online transition distribution and modified transition distribution, i.e., $P_{B_i}$ and $P_{\epsilon_i}$.


* The problem formulation of how to select the best-matched transition online data among the offline data is clear. Considering the distance measurement in terms of both embedding space and latent representation is reasonable.

* Using rank-based prioritization in PER to compute $f$ is reasonable.





**Summary Of The Paper:**

This paper studies the discrepancy between the online and offline transition data in multi-agent RL training. The author proposed OTC to correct the biased transition dynamics and propose a training framework that can leverage offline data for more efficient online training. Results on Mujoco tasks have been provided.

**Summary Of The Review:**

Though the proposed work is sound on the technical side, I feel not surprising about the proposed method. The novelty is rather limited in the sense that the proposed method is a way to shift the offline data for updates in the context of BCQ. The experiment also seems rather limited; showing its effectiveness over BCQ is also not surprising either. Given the novel setting of offline training with online fine-tuning, the author maybe want to consider transfer learning cases like few-shot settings, and experiments on other multi-agent environment such as StarCraftII will also be favoured.

---

> ### Author Response · Authors · 2021-11-15
> **Response to Reviewer ZgL6**
>
> Thanks for your positive review of the technical side of our method. All the reviewers agree that the problem we study is interesting and important. As the first work to study the mismatch between the online and offline transitions in offline MARL, this paper gives a clear problem definition and provides a initial approach to practically address the problem. We do think this work paves the way for further research on this problem. Not superisingly good results happen to be the evident of the problem is hard (with limited online experiences), which leaves the space for further studies. We also provide a theoretical optimization objective in Section 3.2 to minimize the mismatch, however, due to the limited online experiences, it is hard to be optimized by gradient-based method. We sincerely hope you could take this meric into the evaluation. Thank you.

---

> ### Author Response · Authors · 2021-11-24
> **Not surprising effectiveness**
>
> Since OTC corrects the bias between offline and online transition dynamics, it will be more effective when the bias is larger. However, when we split D4RL datasets into individual datasets for each agent, the behavior policy $\pi_{\beta_j}$ in the dataset $i$ is the same as the $\pi_{\beta_j}$ in the dataset $j$. Since agent $j$ is trained using offline RL, the learned policy $\pi_j$ is not far away from $\pi_{\beta_j}$. For agent $i$, the offline transition dynamics, which are determined by $\pi_{\beta_j}$, will not be far away from the online transition dynamics, which are determined by $\pi_{_j}$. That is why the performance is not surprising. In the non-stationary single-agent task (the common concern), where the changes of transition dynamics will greatly influence the performance, OTC could greatly outperform BCQ.

---

### Official Review · Reviewer_a2PP · 2021-11-01

**Correctness:** 3
**Technical Novelty And Significance:** 3
**Empirical Novelty And Significance:** 3
**Recommendation:** 5
**Confidence:** 4

**Main Review:**

The idea that extends deployment efficiency RL to multi agent setting is interesting. Although there are few papers consider offline MARL, they ignore how to consistently improve policy during online deployment. The paper is well-written in most of parts. However, I still have some concerns.

1.	For the high-level idea that only using similar transitions to train the policy, is there any analysis? Intuitively, online exploration aims at finding novel samples that not exist in the offline dataset since these novel samples are helpful to know the environment and build a accurate model. However, the novel samples are not prone to be chosen in the algorithm.
2.	Some details of the transition function are not introduced. How is the transition function represented? Why can the transition probability be modified by the Eq.3? Any deduction or proof? Do agents share one transition probability function?
3.	For implementation, how many transitions are sampled in Line 16 of the Alg. 1? Since all the transitions should be computed for similarity in Line 13, what is the consumption of time?
4.	For the multi-agent setting, do all agents observe the state rather than their own observations? It seems that the method can be directly applied to single-agent setting. Is there any specific design for multi-agent settings?
5.	For baselines, some offline RL methods are ignored like [1]. I understand that these methods are relatively new and may not be published when authors submit the paper. So I just want to notice authors that offline MARL can also be directly used in the setting of the paper. I would not decrease my score due to this reason.

[1] Offline Decentralized Multi-Agent Reinforcement Learning. Jiechuan Jiang and Zongqing Lu. Arxiv 2021


**Summary Of The Paper:**

The paper proposes a novel MARL method to bridge offline training and online fine-turning. More specifically, a transition function is maintained and updated according to the data obtained from online deployment. Similar state-action pairs are used to update the RL method.

**Summary Of The Review:**

Although the paper is interesting, I still have some concerns on multiple aspects. Three important issues are: 1. The details of the algorithms. 2. The unique design for multi-agent settings. 3. The high-level idea of using similar samples for update.

---

> ### Author Response · Authors · 2021-11-15
> **Response to Reviewer a2PP**
>
> > Similar samples and novel samples.
>
> In section 3.4 we have analyzed that using similar transitions could intuitively minimize the KL-divergence between offline and online transition distribution. In our method, online interaction does not aim at exploration but correcting the mismatched transition dynamics in the offline dataset. As the experiences in the offline dataset are diverse enough, novel sample is not the main issue in our settings. However, the transition bias will cause large extrapolation errors, which is the main challenge in offline MARL. In online interaction, the agent acts the learned policy without exploration noise, because if the agents act the exploration policies, the online-collected dataset does not follow the real transition dynamics, since the real transition probabilities depend on the learned policies of other agents.
>
> > Details of the transition function.
>
> The environment takes the joint action $\vec{a}$ and transitions to the next state $s^{\prime}$ according to the transition probabilities.
>
> For Eq.3, when selecting the best-matched transition of  $\left\langle s, a_{i}, s^{\prime}, r\right\rangle$ from D, the probability of selecting $\hat{s}^{\prime}$ is $P_{D_{i}^{k}}\left(\hat{s}^{\prime} \mid s, a_{i}\right)$, and the probability of accepting $\left\langle s, a_{i}, s^{\prime}, r\right\rangle$ given $\hat{s}^{\prime}$ is $f\left(d\left(s^{\prime}, \hat{s}^{\prime}\right)\right)$. Thus the probability of accepting $\left\langle s, a_{i}, s^{\prime}, r\right\rangle$ is $\sum_{\hat{s}^{\prime}} P_{D_{i}^{k}}\left(\hat{s}^{\prime} \mid s, a_{i}\right) f\left(d\left(s^{\prime}, \hat{s}^{\prime}\right)\right)$​, and $Z\left(s, a_{i}\right)$ is a normalization term.
>
> In practice, we do not need to explicitly model the transition function. In Q-learning, modifying the sampling probability by $f$​-function could achieve the same effect of modifying the transition probability.
>
> From the perspective of each agent i, the transition probability is $P_i(s' | s, a_i)=\sum_{a_{-i}} P_{env}(s' | s, \vec{a}) \prod_{j \neq i}^N \pi_j(a_j | s)$, where $P_{env}\left(s^{\prime} \mid s, \vec{a}\right)$ is the same for all agents. However, since other agents' policies $\prod_{j \neq i}^{N} \pi_{j}\left(a_{j} \mid s\right)$ are different, the transition probabilities are different between agents. Here, we follow the settings in MABCQ (Jiang & Lu, 2021).
>
> > How many transitions are sampled in Line 16 of the Alg. 1?
>
> The batch size of updating RL model is 128, which is summarized in Table 3(Appendix).
>
> > Do all agents observe the state rather than their own observations?
>
> Partial observation would cause non-stationarity and transition mismatches. Since we only study the transition mismatches caused by the update of other agents' policies, we let the agents observe the state.
>
> > MABCQ (Jiang & Lu, 2021)
>
> We have cited the settings of MABCQ (Jiang & Lu, 2021) in Introduction, discussed MABCQ in Related Work, and compared MABCQ in Appendix. MABCQ addresses the offline training, but OTC solves the online fine-tuning problem. OTC could be instantiated on MABCQ, which is shown in Appendix.

---

> > ### Comment · Reviewer_a2PP · 2021-11-24
> > **Response to authors**
> >
> > Thanks for response! I have read the response and the revisited paper. Most of my concerns are addressed. However, the lack of  unique design for multi-agent setting still exists. I sugguest authors to also try the method in single agent setting and give more analysis about the similar samples. The connection between Eq.3 and Eq.8 is also not very clear and more analysis should be done to show that Eq.8 is a good  approximation of Eq.3.

---

> > > ### Author Response · Authors · 2021-11-24
> > > **Response**
> > >
> > > Thanks for your comments! OTC is uniquely designed for the mismatches between offline and online transition dynamics, including non-stationary single-agent settings and decentralized multi-agent settings. However, most single-agent environments are stationary, but non-stationarity is a common issue in all decentralized multi-agent environments. That is why we choose multi-agent settings.
> > >
> > > We test OTC on a non-stationary sparse-reward single-agent task, where an agent learns to reach a landmark. The speed of the agent is $\beta a$, where $a$ is the action and $\beta$ is a random variable. During the data collection,  $\beta$ is uniformly chosen from $[-0.2,0.2]$, which makes the expected movement of each action be $0$. Thus Q values learned from offline dataset are all equal to $0$, and the agent cannot finish this task. During online fine-tuning, $\beta$ gradually changes from $0.1$ to $0.2$. The results of each deployment are recorded in Table 3. OTC outperforms BCQ, which shows OTC could help the fine-tuning in non-stationary single-agent settings.
> > >
> > > Table 3: Mean success rate of each deployment under five different seeds.
> > >
> > > | Deployment | 1    | 2      | 3    | 4      | 5      | 6      | 7      | 8      | 9      | 10     |
> > > | ---------- | ---- | ------ | ---- | ------ | ------ | ------ | ------ | ------ | ------ | ------ |
> > > | $d_e$+BCQ  | $0$​  | $0 $​   | $0$​  | $0.04$​ | $0.55$​ | $0.99$​ | $0.98$​ | $1$​    | $1$    | $1$    |
> > > | BCQ        | $0$​  | $0.06$​ | $0$​  | $0.06$ | $0.04$ | $0.21$ | $0.79$ | $0.70$ | $0.41$ | $0.68$ |
> > >
> > > From Eq.3, we obtain an optimization objective of $f$-function. Due to the limitation of online experiences, it is hard to solve the optimal $f$-function by gradient-based optimization. Thus, we use Eq.8 as an empirical substitute of $f$​​-function, rather than the approximation of Eq.3. With the same state-action pair, if the next state of the offline transition is more similar to the next state of the online transition, the offline transition is more likely to follow the online transition dynamics and should be given a larger acceptance probability. Eq.8 satisfies the monotonicity mentioned above, so we believe it is a good choice.

---

### Official Review · Reviewer_2xuS · 2021-11-01

**Correctness:** 3
**Technical Novelty And Significance:** 2
**Empirical Novelty And Significance:** 2
**Recommendation:** 3
**Confidence:** 5

**Main Review:**

Novelty: The bias of transition dynamics in offline decentralized MARL problem is interesting.

Soundness:
1.This paper studies the deployment constrained setting. However, BREMEN and MUSBO are never compared in the experiments. It is obvious that modifying the transitions given the new data improves the learning performance.
2.It seems that the OTC technique can also be applied in the single agent setting. Does OTC performs better in MARL than SARL?
3.In each iteration, OTC needs to find the most similar samples for each sample in the mini-batch, which may incur large computation cost.

Significance:
The performance improvement of OTC compared with BCQ is convincing. But baselines such as BREMEN are missing.



**Summary Of The Paper:**

This paper studies the problem of online tuning for offline decentralized MARL. The online transition correction technique is proposed to correct the bias of the transition probability function between offline dataset and online dataset. Experimental results show that OTC is able to improve the performance of BCQ and AWAC given the same number of deployments and samples.

**Summary Of The Review:**

This paper studies an interesting problem in offline MARL. A transition modification technique, OTC is proposed to correct the transition bias during the learning. However, the experimental evaluation is not convincing as important baselines are missing, and the proposed algorithm seems to be too straightforward due to lack of theoretical analysis. Thus, I recommend the rejection.

---

> ### Author Response · Authors · 2021-11-15
> **Response to Reviewer 2xuS**
>
> > BREMEN and MUSBO
>
> Although BREMEN and MUSBO consider the deployment efficiency, the problem settings of BREMEN and MUSBO are totally different from the settings of OTC. OTC fine-tunes the policies pre-trained from the offline dataset by limited deployments. BREMEN and MUSBO do not fine-tune the pre-trained policies but start from an empty dataset and train the policies from the scratch every deployment. Thus BREMEN and MUSBO are not suitable in our settings and achieve bad performance, as shown in Table 2.
>
> Table 2: Performance of BREMEN
>
> |                    | OTC($d_e$)    | BREMEN       |
> | ------------------ | ------------- | ------------ |
> | halfcheetah-random | $1242 \pm 11$ | $987 \pm 23$ |
> | walker2d-random    | $446\pm24$    | $315\pm14$   |

---

> ### Author Response · Authors · 2021-11-24
> **Looking forward to further comments**
>
> Thanks for your helper comments! We reported the computation cost and analyzed the single agent settings in common concern. If you have any other questions, please let us know. We are looking forward to having further discussion.

---

### Official Review · Reviewer_T8NR · 2021-11-04

**Correctness:** 2
**Technical Novelty And Significance:** 2
**Empirical Novelty And Significance:** 2
**Recommendation:** 5
**Confidence:** 4

**Main Review:**

- Pros:

(1) Authors study an interesting and practical setting, offline training and online tuning for multi-agent tasks.

(2) The paper is well-written, with a clear description of the proposed method.

(3) Authors conduct extensive experiments to evaluate OTC.

- Cons:

(1) Related work: The idea of correcting sampling probabilities is very close to that in (Jiang and Lu, 2021), and authors should discuss more about MABCQ in the paper.

(2) Method:
The method involves a pairwise comparison of the sampled transition and all transitions from online execution, which can incur some computation overheads when D_i^k becomes larger. In addition, the paper also assumes there is always <s^*, a_i^*, s^{'*}, r^*> in the dataset, which may not always be the case. In the d-function part (Equation 7), it leverages the difference in value functions for measuring the distance. However, Q(s, a) can be over-optimistic for unseen actions.

(3) Experiments:

- There is a missing strong baseline MA-ICQ (Yang et al., 2021) which also studies the offline MARL setting.

- Given Figure 4, there seems to be very large variance, and I'm not very confident about the conclusion.

 - In most experiment tasks, there only involves 2 agent, which make it relatively small scale. Can the method scale with more agents?


**Summary Of The Paper:**

The paper studies offline training and online tuning for multi-agent systems, which focuses on the decentralized cooperative setting where each could only access to its offline dataset. Authors discover that the transition dynamics in the offline dataset and online execution can differ very much, and propose to bridge this gap by introducing online transition correction by modifying the sampling probabilities. To accomplish this goal, authors propose practical implementations for the distance function measuring the similarities of states/state-action pairs. Authors conduct extensive experiments on multi-agent MuJoCo tasks, and show that OTC outperforms baselines (BCQ and AWAC).

**Summary Of The Review:**

The paper studies an interesting setting, but should explain more about the potential problem of the method (please check the main review) and compare it with a more recent MA-ICQ method.

---

> ### Author Response · Authors · 2021-11-15
> **Response to Reviewer T8NR**
>
> > MABCQ (Jiang & Lu, 2021) and MA-ICQ (Yang et al., 2021)
>
> We have discussed MABCQ (Jiang & Lu, 2021) in Section 2.1 and compared MABCQ in Appendix. MABCQ addresses the offline training but OTC solves the online fine-tuning problem. OTC could be instantiated on MABCQ, which is empirically shown in Appendix. Since the transition correction in MABCQ is not based on online transition dynamics, the modified transition dynamics in MABCQ are not necessarily close to the online ones.
>
> MA-ICQ is a centralized-training method that requires joint actions, but we adopt the setting of fully decentralized training, where the joint actions is **not** available. So MA-ICQ cannot be applied to decentralized datasets. We will include the discussion about MA-ICQ in the revision.
>
> > Assumption
>
> Assuming that there is always $<s^*, a_i^*, s', r^*>$ in the dataset is only for the convenience of theoretical analysis. In Section 3.3 we have discussed that it is impossible to always find the transition from $\mathcal{D}_{i}$ with the same state-action pair as the given transition. But we could also estimate the transition similarity relying on the generalization ability.
>
> > Scaleability
>
> OTC is a decentralized method that takes other agents as a part of the environment regardless of the number of agents and does not require any information of other agents. Thus OTC could scale well in large-scale environments.

---

### Author Response · Authors · 2021-11-15
**Common Concern**

> Straightforward method, no theoretical analysis,  not suprisingly good performance

All the reviewers agree that the problem we study is interesting and important. As the first work to study the mismatch between the online and offline transitions in offline MARL, this paper gives a clear problem definition and provides a initial approach to practically address the problem. We do think this work paves the way for further research on this problem. Not superisingly good results happen to be the evident of the problem is hard (with limited online experiences), which leaves the space for further studies. We also provide a theoretical optimization objective in Section 3.2 to minimize the mismatch, however, due to the limited online experiences, it is hard to be optimized by gradient-based method. We hope the reviewers could take this meric into their evaluations.


> Computation cost

For the computation cost, in section 3.5 we have stated that to reduce the complexity, for each update we uniformly sample two minibatches, B and D respectively from $B_{i}^{k}$ and $D_{i}^{k}$, and perform the rank-based prioritized sampling on the two minibatches. The computation of transition similarity is on GPU, which is fully parallelized and very efficient. When GPU memory and CUDA cores are enough, the cost of computing 1 sample is almost equal to the cost of computing N samples. In Table 1, we record the average time taken by one update. The computation of transition similarity in OTC costs less than 25% of the BCQ training time. The experiments are carried out on Intel i7-8700 CPU and NVIDIA GTX 1080Ti GPU.

Table 1: Average time taken by one update.

| BCQ    | $d_e$+BCQ | $d_q$+BCQ |
| ------ | --------- | --------- |
| 11.6ms | 14.2ms    | 13.8ms    |


> Could OTC be applied in the single-agent settings? Is there any specific design for multi-agent settings?

Since OTC deals with the mismatches between the offline transition dynamics and online transition dynamics, it could be applied in both non-stationary single-agent settings and decentralized multi-agent settings.

In most stationary single-agent tasks, the transition dynamics do not change, and the offline transition dynamics follow the online transition dynamics, so OTC cannot bring performance gain in such cases. In non-stationary single-agent settings[1,2,3], where the transition probabilities are changing, the offline transition dynamics are different from the online transition dynamics, thus OTC could work. As pointed by [1], decentralized MARL is an important case of non-stationary settings,  where non-stationarity is an inherent issue. Since other agents are updating their policies,  the offline transition dynamics are usually much different from the online transition dynamics, causing large extrapolation errors. OTC could reduce the transition bias and minimize the extrapolation error in MARL.

[1] Near-Optimal Model-Free Reinforcement Learning in Non-Stationary Episodic MDPs

[2] Optimizing for the Future in Non-Stationary MDPs

[3] Towards Safe Policy Improvement for Non-Stationary MDPs

---

### Author Response · Authors · 2021-11-24
**Single agent task**

In Common Concern, we have analyzed that OTC could be applied in both non-stationary single-agent settings and decentralized multi-agent settings. We test OTC on a non-stationary sparse-reward single-agent task, where an agent learns to reach a landmark. The speed of the agent is $\beta a$, where $a$ is the action and $\beta$ is a random variable. During the data collection,  $\beta$ is uniformly chosen from $[-0.2,0.2]$, which makes the expected movement of each action be $0$. Thus Q values learned from offline dataset are all equal to $0$, and the agent cannot finish this task. During online fine-tuning, $\beta$ gradually changes from $0.1$ to $0.2$. The results of each deployment are recorded in Table 4. OTC outperforms BCQ, which shows OTC could help the fine-tuning in non-stationary single-agent settings.

Table 4: Mean success rate of each deployment under five different seeds.

| Deployment | 1    | 2      | 3    | 4      | 5      | 6      | 7      | 8      | 9      | 10     |
| ---------- | ---- | ------ | ---- | ------ | ------ | ------ | ------ | ------ | ------ | ------ |
| $d_e$+BCQ  | $0$​  | $0 $​   | $0$​  | $0.04$​ | $0.55$​ | $0.99$​ | $0.98$​ | $1$​    | $1$    | $1$    |
| BCQ        | $0$​  | $0.06$​ | $0$​  | $0.06$ | $0.04$ | $0.21$ | $0.79$ | $0.70$ | $0.41$ | $0.68$ |

---

### Decision · Program_Chairs · 2022-01-20

**Decision:**

Reject

**Comment:**

## A Brief Summary of the Paper

 In offline decentralized MARL, the discrepancy between the offline data and the interactions of agents in the environment can cause discrepancy and as a result, the policies will perform suboptimally. This issue in ORL is known as extrapolation error. This paper is trying to address the issue of extrapolation error with offline decentralized agents. It is possible to alleviate this problem by combining offline RL with online fine-tuning. This paper introduces Online Transition Correction (OTC) approach to address this problem which aims to correct the biased transition dynamics with a form of importance sampling based on embedding and value-based distance metrics.

 ## Summary of the Reviews

 Below I will outline some important concerns raised by the reviewers.

 ### Reviewer T8NR
 **Pros:**
 - Interesting and practical setting.
 - Extensive experiments to evaluate OTC.
 **Cons:**
 - Lack of enough discussions about closely related works, for example, the MABCQ algorithm.
 - Computational cost of OTC.
 - Experiments: comparisons against baselines (MA-ICQ), Large variance in Figure 4. Small-scale (only two agent) setting, can it scale to more agents?

### Reviewer 2xuS

 **Pros:**
 - The bias of transition dynamics in offline decentralized MARL problem is interesting.
 **Cons**
 - Key baselines such as [[BREMEN]] and [[MUSBO]] that looks into the deployment constrained ORL is not compared against in this paper.
 - The proposed OTC algorithm can easily be applied to Single agent settings. MARL vs SARL comparisons would be interesting.
 - Large computational cost incurred from OTC. Because of the search procedure of finding most similar examples to the examples in the dataset.

### Reviewer a2PP
**Pros:**
- Well-written.

**Cons:**
- Analysis on the behavior of the policy, in particular on novelty seeking during the online finetuning phase.
- The missing details of the transition function.
- Compute constraints and budget.
- Unclear experimental protocol: states vs observations...
- Missing baselines.

### Reviewer ZgL6
**Cons**
- Lack of novelty.
- Limited experimental results: transfer learning scenarios, lack of experiments on multiagent environments such as Starcraft II.

## Key Takeaways and Thoughts

Overall, the authors did a good job addressing the concerns raised by the reviewers. For example, the authors ran additional experiments and compared single-agent BCQ with and without OTC on some D4RL tasks. The authors gave detailed responses to the questions related to the computational cost. However, the initial submission version of this paper feels rushed as it is submitted to the conference. I would recommend the authors, go through the reviews carefully and address the points raised by the reviewers carefully in a future version of this paper. As it stands now, it is difficult to evaluate the results reported by the authors during the rebuttal, due to the lack of clarity about their experimental details.

I think the writing can be improved further. There are several typos in the paper, and most reviewers were confused about the novelty of the paper. I would recommend the authors to provide more detailed discussions about the differences from other similar approaches in the literature. Also, justify the selected experimental protocol better.